# Influence of Mediterranean Diet on Human Gut Microbiota

**DOI:** 10.3390/nu13010007

**Published:** 2020-12-22

**Authors:** Giuseppe Merra, Annalisa Noce, Giulia Marrone, Marco Cintoni, Maria Grazia Tarsitano, Annunziata Capacci, Antonino De Lorenzo

**Affiliations:** 1Section of Clinical Nutrition and Nutrigenomic, Department of Biomedicine and Prevention, University of Rome Tor Vergata, via Montpellier 1, 00133 Rome, Italy; marco.cintoni@gmail.com (M.C.); delorenzo@uniroma2.it (A.D.L.); 2UOC of Internal Medicine, Center of Hypertension and Nephrology Unit, Department of Systems Medicine, University of Rome Tor Vergata, via Montpellier 1, 00133 Rome, Italy; annalisa.noce@uniroma2.it (A.N.); giul.marr@gmail.com (G.M.); 3PhD School of Applied Medical, Surgical Sciences, University of Rome Tor Vergata, via Montpellier 1, 00133 Rome, Italy; 4Department of Experimental Medicine, Sapienza University of Rome, Viale Regina Margherita 324, 00100 Rome, Italy; mariagrazia.tarsitano@gmail.com; 5Department of Gastroenterological, Endocrine-Metabolic and Nephro-Urological Sciences, “Agostino Gemelli” General Hospital Foundation-IRCCS, Largo A. Gemelli 8, 00168 Rome, Italy; annunziata.capacci@guest.policlinicogemelli.it

**Keywords:** Mediterranean diet, gut microbiota, polyphenols, fiber, ω-3 PUFA

## Abstract

Gut microbiota changes correlate with health status. Literature data on gut microbiota show that all dietary changes can induce the alteration of gut microbiota composition. Mediterranean diet (MD) is associated with a reduction of all-cause mortality and in this review, we analyzed its interactions with human microbiota. In particular, we explored the modulation of the human microbiota, in response to MD adherence, focusing the attention on polyphenols, polyunsaturated fatty acids (PUFA) ω-3 and fiber. Evidences suggest that MD is able to modulate the gut microbiota, increasing its diversity. In fact, a Mediterranean-type dietary pattern is associated with specific gut microbiota characteristics. The available evidence, suggests that gut microbiota of subjects that follow a MD is significantly different from subjects that follow a Western diet model. In fact, the latter show an increased gut permeability, which is responsible for metabolic endotoxemia. For this reason, we can speculate that the gut microbiota of the subjects following a MD is able to prevent the onset of chronic non-communicable degenerative diseases, such as cardiovascular diseases and some types of cancer. However, in order to understand these correlations with dietary patterns, controlled intervention studies on the gut microbiota composition and activity are needed.

## 1. Introduction

The “concept” of Mediterranean diet (MD) [1] was introduced and characterized for the first time by Ancel Keys, a physiologist, who subsequently studied the effects of eating habits on the incidence of cardiovascular (CV) diseases. The results of Keys work have produced the well-known Seven Countries Study [2,3]. The Seven County Study was an epidemiological study, which aimed to evaluate the relationship between diet and human health, in seven countries like United States, Finland, Netherlands, Italy, Yugoslavia, Japan and Greece (Crete).

The results of this study clearly showed that the Cretan sample was by far the least involved in CVD, despite moderate to high fat consumption (most of which represented by olive oil). One of the main conclusions of the study was that CVD could be prevented and were strongly influenced by the dietary fat composition. Many subsequent studies confirmed these results. The Lyon Diet Heart Study, published in the 1990s, also pointed this out on 605 people, who had all recently had a myocardial infarction, were divided into two groups. The first group followed a classic diet recommended for these diseases, instead the second group followed a Mediterranean food style abounding in rapeseed oil (rich in ω-3). This study showed a clear drop in mortality (−70%) in the group following a MD, in fact, due to this detectable effect attributable to dietary intervention, the study was stopped prematurely [4,5,6]. 

The PREDIMED study [7], with 7447 participants, then, was the largest interventional study carried out on a MD and confirmed the CVDs prevention [8], the increase in life expectancy and also the reduction of the Alzheimer’s disease onset [9,10].

The European Prospective Investigation into Cancer and Nutrition (EPIC) study was conducted on 519,978 subjects, from 10 European countries (Denmark, France, Germany, Greece, Italy, Holland, Norway, Spain, Sweden and the United Kingdom). The purpose of the EPIC study was to investigate the relationships between diet, environmental and lifestyle factors, and the incidence of cancer and other chronic diseases. All study participants were contacted on a regular basis, at 3–5 years intervals, to evaluate any changes in their lifestyle. The obtained data showed that MD represents the best food model in the prevention of cancer diseases; moreover, the intake of flavonoids (content blueberries, strawberries, grapes, melons, citrus fruits, apricots, onions, cabbage, fennel, tomatoes, lettuce, broccoli, spinach, etc.) would seem to reduce the risk of developing gastric cancer [11,12].

The Italian cohort of the Epic study highlighted how the food model “Olive Oil and Salad”, characterized mainly by the consumption of raw vegetables, extra virgin olive oil (EVOO) and legumes, is related with a minor mortality risk in geriatric patients and a lower incidence of colorectal cancer. As for colorectal cancer, regular yogurt consumption is also associated with a reduction of its incidence, likely related to the beneficial action exercised by probiotics [13,14].

The traditional MD [15] (Figure 1) has its origin in the countries of the Mediterranean basin [16], where the sunny and mild climate favors the production of a lot of fruit and vegetables throughout the year. Therefore, it is characterized by the consumption of fruit and vegetables in abundance, by using EVOO as the main source of fat, and by the consumption of legumes, whole grains, nuts, seeds and aromatic herbs.

Moreover, it provides moderate consumption of dairy products (mainly in the form of fermented products such as yogurt and cheese), eggs, fish and red wine in small amount during meals. Meat consumption is often reduced (once a week on average), favoring lean meats (rabbit, chicken and turkey) [17].

Meals are always accompanied by ingredients that give flavor and contain substances with health benefits. First, onion and garlic (two very rich food of prebiotics that make up the “sauté” [17], which is the basis of many Mediterranean cuisines), as well as fresh or dried aromatic herbs (parsley, oregano, mint, rosemary, thyme, coriander, basil) and spices (cumin, cloves, saffron, cinnamon, pepper…), which bring numerous antioxidant and anti-inflammatory components [18].

A higher proportion of plant-based proteins (which mainly come from legumes and whole grains), than the animal proteins, contributes to increase the beneficial effects of MD [19].

The MD also seems to represent an alternative nutritional approach respect to the low-protein diet in chronic kidney disease (CKD) patients (stage II and III according to the K-DIGO guidelines) [20]. In fact, in our previous study, we showed that 14 days of Italian MD (IMD), followed by 14 days of Italian Mediterranean Organic Diet (IMOD) appear to reduce homocysteine values and improve body composition in patients with CKD under conservative therapy. These effects were also influenced by the genotype of the methylenetetrahydrofolate reductase (MTHFR) [19].

Our subsequent study conducted on CKD patients, demonstrated that IMOD, according to “Nicotera model”, induced a decrease in homocysteine, serum phosphorus and albuminuria values and in fat mass both in percentage and in kg [15].

Among the characteristic foods of the MD, one that is known to induce health benefits is EVOO. The latter is rich in monounsaturated fatty acids, that are beneficial for the heart health and rich in polyphenols, which are antioxidants and they belong to the same family of red wine tannins [21].

The role of diet in the gut microbiota modulation has been widely recognized [22,23]. De Filippis et al. [24] demonstrated that a Mediterranean-type diet, characterized by an high content of plant-based food, exert a beneficial role in the gut microbiota composition. In particular, subjects who eat a higher proportion of plant-based foods show a higher percentage of short chain fatty acids (SCFAs) and of fiber-degrading bacteria in the feces. While, subjects with poor adherence to the MD have a higher concentration of trimethylamine N-oxide (TMAO) in the urine. 

In a subsequent study Mitsou et al. [25] confirmed the positive impact of the MD on the gut microbiota profile. In fact, subjects who had a higher adherence to the MD, monitored with the MedDietScore, had a lower presence of *E. coli* and an increased total presence of bacteria, an higher Bifidobacteria/*E. coli* ratio and an increased prevalence of *C. albicans* and SCFAs.

Recently, Garcia-Mantrana et al. [26] pointed out that an higher adherence to the MD is characterized by an enhancement of Bifidobacteria and a higher percentage of SCFAs. Therefore, MD has a positive impact on gut microbiota, in particular on its α-diversity and metabolic activities.

In this review, we analyzed the impact of MD on gut microbiota composition. In fact, we explore the possible modulation of the gut microbiota, in response to MD adherence, focusing the attention on MD typical elements such as EVOO, polyunsaturated fatty acids (PUFA) ω-3 and fiber.

## 2. Materials and Methods

We conducted a literature search on PubMed, electronic database using the keywords “Mediterranean diet” [Title/Abstract] and “gut microbiota” [Title/Abstract] or “gut microbiome” [Title/Abstract] or “extra-virgin olive oil” [Title/Abstract] or “polyunsaturated fatty acids” [Title/Abstract] or “fiber” [Title/Abstract].

Reference lists and related records were manually reviewed. The search was limited to English language papers published until August 2020.

## 3. Mediterranean Diet and Microbiota Composition

Although the concept of microbiota is not yet well-known, the human intestine is colonized by millions of bacteria that contribute to its formation. The composition of the microbiota deeply influences the human health and the diet plays a pivotal role on its composition.

The interaction between diet and gut microbiota is mutual. While, the microbiota acts on digested nutrients, the influence of food has a strong impact on the intestinal microbial composition. Metabolic activities of the latter depend largely on the amount of non-digestible carbohydrates and proteins that reach the intestine. Animal and human studies of gut microbiota fecal samples showed that all dietary changes can induce a modulation of gut microbial composition. In healthy subjects, a balanced diet can ensure the formation of a good microbial flora, where the content of all species of bacteria live in a system of control but also of mutual balance [27].

Through multiple studies conducted on the correlation between diet and gut microbiota, it has been shown that a diet rich in fat, with high consumption of red meat and refined carbohydrates, and poor in fish, plant-based foods and fruit, may have a direct effect on the immune system causing dysbiosis. Dysbiosis is a structural and functional modification of the gut microbiota, capable of triggering severe inflammation, through a greater number of pro-inflammatory micro-organisms resulting from a reduction in diners, promoting mechanisms of immune tolerance [28]. The gut microbiota reacts quickly to dietary changes. In fact, it has been proposed that the diet is able to change almost 60% of its total structure [29]. In the gut, the dominant species are represented by the phyla Bacteroidetes, Firmicutes and Actinobacteria [30].

A diet that includes the consumption of animal proteins and fats is linked to the enterotype dominated by Bacteroides, on the contrary, a diet rich in carbohydrates is associated with the enterotype dominated by Prevotella. The major phylum present in the gut microbiota are Firmicutes and Bacteroidetes, which represent about 90% of the intestinal bacterial flora [31].

The main genera that make up the Firmicutes phylum are Clostridium, Enterococcus, Lactobacillus and Ruminococcus. While the main genera of the phylum Bacteroidetes are Prevotella and Bacteroides. Many pathological conditions are characterized by a dysbiosis of the intestinal microbiota [14]. For example, in presence of CKD, the alteration of the gut microbiota composition is characterized by the prevalence of Enterococcaceae, Lachnospiraceae, Ruminococcaceae families and a decrease of Bifidobacteriaceae, Baceroidaceae, Lactobacillacee and Prevotellaceae families [14]. Moreover, it has recently been shown that dysbiosis is directly correlated with the onset of colorectal cancer (CRC). In particular, the presence of some bacterial species such as *Fusobacterium nucleatum*, *Peptostreptococcus anaerobius* and enterotoxigenic, *Bacteroides fragilis* can cause the proliferation of tumor cells, induce an inflammatory state and damage to the DNA with consequent impact on the onset of CRC. While, the reduction of probiotic bacteria such as *Lachnospiraceae species*, *Bifidobacterium animalis* and *Streptococcus thermophilus*, that seem to exert a protective action in CRC, is present in patients affected by this neoplastic pathology [32].

The gut microbiota composition is strongly influenced by probiotics consumption (live microorganisms) that, due to their resistance, reach the colon temporarily and grow becoming metabolically active, but also through the use of prebiotics (compounds, often fiber, that induce the growth of beneficial bacteria species) [33].

It has long been known that the microbial species promotes the growth and development of intestinal epithelial cells that are involved in the metabolism of important nutrients, such as carbohydrates, proteins and fats [34].

At the base of human intestinal flora imbalance there are several causes, such as stress, infections, but especially the wrong eating habits. Therefore, if the balance fails, the intestinal microbiota can be altered in type, quality and structure [35].

In general, when a lowering between Firmicutes and Bacteroidetes ratio is observed, a microbial flora producing SCFA prevails, rather than TMAO. This condition is typical of Western-diet model (Figure 2).

The MD is characterized by an high amount of dietary fiber, and also of polyphenols (which would exert a prebiotic action on specific strains) [21,36]. Due to the high production of SCFA (butyrate species), induced by MD, the microbiota in the subjects who follow MD, seem to contribute to incidence reduction of the of some type of cancers (especially the CRC) or of the cardio-metabolic pathologies [37,38]. The protective health effects associated with the consumption of a Mediterranean-type diet has in scientific literature long been recognized (Figure 3).

The combination of the dietary components typical of the MD and the microbiota composition leads to the production of specific metabolites, such as SCFA (represented in the feces of subjects that follow MD), conversely TMAO metabolites are present in higher concentrations in subjects that follow a Western-type diet [24]. Studies of modification of the dietary pattern unfortunately show that it is not easy, at least in the short-term, in order to influence significantly, and above all stabilize, the intestinal microbiota. In a randomized controlled trial, Djuric explored the mucosal bacterial flora of the colon before or after six months of the Mediterranean or Western type experimental diet, without observing significant differences in the pre- or post-intervention microbiota [39]. In a study conducted in patients with metabolic syndrome fed with a Mediterranean or traditional diet for two years, the MD has instead shown that it could reduce, although only in part, the dysbiosis typical of the metabolic syndrome. In fact the authors observed an increase of Bifidobacterium genera [37] in the MD group [19], that seem to have a possible anticancer action. Their altered metabolism in neoplastic cells, would activate phenomena (such as histone acetylation) able to induce apoptosis [38]. The highest levels of SCFA, and species of butyrate, at colon-rectal level would contribute in particular to explaining the reduced risk of CRC observed in the Mediterranean world. This protective effect could also be attributable to the reduced presence of *Fusobacterium nucleatum*, often present in the colon of patients with CRC, and according to some authors perhaps causally related to its onset [40]. It is also documented that the levels of this bacterium increase in the colon just two weeks after switching to a Western-style diet [41]. Defining the causality of these associations is naturally complex. In an experimental mouse model, the use of a Mediterranean dietary mix significantly reduced the incidence of colon cancers associated with the treatment with azimethane, apparently due to the selective specific modification of the microbiota induced by the dietary mix [42].

Recently, Tosti et al. [9] investigated if the typical health benefits induced by the MD could be attributable from the specific microbiota associated with it. This hypothesis is beginning to gain support in the literature, even if studies that have analyzed the correlation between MD and the gut microbiota composition are scarce. Furthermore, most of these studies are transversal, which means that they have detected the characteristics of the microbiota in subjects with different degree of adherence to the MD model; in fact, these studies unfortunately are not useful to define the presence of cause-effect relationships. For this reason it is important to understand whether the effects induced by dietary patterns act on the change in the gut microbiota composition and therefore on the probability of disease onset, or if these effects are exclusively attributable to diet-induced changes in the gut microbiota. Currently, it is estimated that more than half of the gut microbiota variability is attributable to the diet.

In most of the published studies, in fact, the consumption of a Mediterranean-type diet is associated with a different microbiota from that associated with a Western-type dietary pattern. The microbiota, that could be brought back to the MD type, is first of all characterized, by a greater biodiversity (i.e. by a greater number of identified bacterial species), a feature that has a positive impact on human health. This feature of gut microbiota is defined “α-diversity” as it expresses the number of species present in the microbiota and is associated with the subject state of health [43]. More specifically, the Western diet is associated with high levels of Bacteroides, while the genus Prevotella is more represented in the MD. Gutierrez-Diaz et al. identified higher levels of Clostridium of cluster XIVa and *Faecalibacterium Prausnitzii* in subjects with a high score of adherences to the MD (MDS score > 4) [44]. The same score, in another publication of the same group, was associated with a greater abundance of Bacteroidetes, Prevotellaceae and Prevotella and a lower presence of Firmicutes and Lachnospiraceae [45]. Also, Garcia-Mantrana et al. documented a lower Firmicutes-Bacteroidetes ratio associated with a high adherence to the MD. Typical food components present in the MD are associated with the presence of specific strains in the gut microbiota. For example, cereals consumption are related to the presence of Bifidobacterium and Faecalibacterium, Tenericutes and Dorea. The consumption of olive oil and red wine are related to the presence of Faecalibacterium, the consumption of vegetables are related to the presence of Rikenellaceae, Dorea, Alistipes and Ruminococcus, legumes are related to the presence of Coprococcus species [46].The same authors have also observed a correlation between the polyphenol content of the diet (typically high in the MD) and the presence of specific clostridium (XIVa) and Faecalibacterium clusters, capable of synthesizing butyrate and probably endowed with an action anti-inflammatory (like Akkermansia, also more represented in association with the MD) [26].

### 3.1. Extra-Virgin Olive Oil

Bioactive phytochemicals are well-known for inducing beneficial effects on human health. The most phytochemical compounds represented in the MD are polyphenols, the secondary metabolites of plants, derived from phenylalanine and tyrosine with phenolic base structure [47].

The cornerstone food of MD is EVOO which is particularly rich in secondary phenolic compounds, as widely demonstrated in the literature [48,49]. EVOO owes beneficial properties due to oleic acid and polyphenols. In fact, these substances have an high antioxidant action which in turn counteracts oxidative damage [50,51]. The antioxidant and anti-inflammatory action performed by polyphenols can be carried out through various mechanisms. Among these, we find a direct action in the bowel (where they are absorbed), or an action linked to the transformation in the gastrointestinal tract, that produces bioactive metabolites able to modulate biological responses [52].

An *in vitro* study conducted by Romero C et al. demonstrated that polyphenols present in olive oil can spread in gastric juice, survive for several hours in the acidic environment, and exert a significant bactericidal effect against eight strains (three of which resistant to some types of antibiotics) of *Helicobacter pylori* at very low concentrations (1.3 μg/mL). Such *in vitro* experimental data paved the way for the possibility to use a simple food, such as EVOO, as a chemopreventive agent for ulcer and/or gastric cancer, and therefore, its potential use should be tested in *in vivo* studies [53].

Another important action performed by the microbiota is represented by immuno-modulation. Martín-Peláez et al. observed, in 10 hypercholesterolemic subjects, a change in the intestinal mucosa following the intake of enriched virgin olive oil with 500 mg of phenolic compounds/kg content. The results obtained, showed a stimulation of the immune system (through monitoring of fecal immunoglobulins-A coated bacteria and plasma C reactive protein). These data highlight the link between minor polar compounds present in olive oil and the host response, modulated through the microbiota [54].

An *in vivo* 3-months study on 18 overweight/obese subjects *vs* 18 normal weight subjects (control group) has shown that MD, that included 40 g per day of EVOO is able to modulate the composition of the gut microbiota, specifically lactic acid bacteria. The authors showed that the consumption of EVOO is able to improve the anti-inflammatory action of the MD through the reduction of myeloperoxidase (marker of inflammation and endothelial dysfunction), 8-hydroxy-2-deoxy-guanosine (oxidative DNA damage marker), tumor necrosis factor (TNF)-α and interleukin (IL)-6 (cytokines involved in the inflammatory response). In addition, these authors demonstrated an increase in adiponectin (protein which regulates gene expression of IL-10) at the end of the study [55].

### 3.2. Polyunsaturated Fatty Actids ω-3

The MD is known to contain an high amount of blue fish. This food is very rich in polyunsaturated fatty acids, known to most as “PUFA”, and over the years, many authors have shown their beneficial effects in reducing cardiovascular events in those who used them [56,57,58,59].

The most studied PUFAs, able to reduce the inflammation status, are ω-3. In particular, the main bioactive forms capable to confer healthy effects in the human organism are: (1) eicosapentaenoic acid (EPA, C20: 5), (2) docosahexaenoic acid (DHA, C22: 6) and (3) α-linolenic acid (ALA, C18:3) [60]. Foods as fish, seafoods and nuts are rich in ω-3 fatty acids, which contribute to the good balance between ω-6/ω-3 and consequently to decrease inflammation [61].

In a diabetic rat model study [62], the administration of flaxseed oil (rich in α-linoleic acid) seems to improve glucose metabolism, reduce interleukin 1-β, TNF-α and malondialdehyde, compared to the control group. In particular, the analytical sequencing of the intestinal microbiota showed a reduction of Firmicutes and *Blautia* that positively correlate with inflammatory cytokines such as interleukin-1β, TNF-α, LPS, etc. Dietary intake with linseed oil appears to improve glycometabolic control in type 2 diabetes mellitus by suppressing inflammation through modulation of the intestinal microbiota.

Moreover, beyond the well-known cardioprotective properties, several authors, in the last decade, investigated the interaction between ω-3 dietary consumption and gut microbiota composition in order to identify changes in the composition of microbial species resident in the host. The role of ω-3 on the gut microbiota would seem to modulate the inflammatory response which lies at the base of several chronic non-communicable degenerative diseases (CNCDDs), such as atherosclerosis, cancer, neurodegenerative diseases, chronic renal failure, diabetes mellitus, male obesity secondary hypogonadism etc. [63,64,65,66]. Moreover, several studies speculate that the microbiota-PUFA ω-3 interaction is capable of modulating the immune system and inflammatory status [60,67]. In this regard, *in vivo* animal studies have investigated intestinal permeability, highlighting that ω-3 PUFAs improve the epithelial barrier in the presence of colitis [68].

### 3.3. Fiber

The dietary fiber plays a prebiotic action for the microbial growth of bacteria living in the human intestine. The classification of dietary fibers is based on monomer units (MU), and it allows to distinguish them into polysaccharides (consisting of at least 10 MU) or oligosaccharides (consisting of 3 to 9 MU). Their further classification is based on their aqueous solubility, viscosity and fermentability [69].

Oligosaccharides, beta-glucans, gums and some celluloses, therefore in general fermentable fibers, are an excellent substrate for the metabolism of bacteria to produce SCFAs, and in particular, acetate, propionate and butyrate [70]. These metabolites, as reported in recent studies, play an important roles in regulating immunity, blood pressure, glucose and lipid metabolism, representing the link between the microbiota and the human homeostasis [67].

Among dietary patterns, the MD provides a minimum of 14 g of fiber for every 1000 kcal, therefore it provides at least daily double amount allowance of fiber, compared to the Western diet. These consumption of fibers decrease cholesterol and insulin levels and improve gut microbiota composition [71,72].

An intake of fiber (20 g of inulin) in overweight and obese subjects would seem to improve the α-diversity of the gut microbiota and increase the “bifidogenic effect” [73].

A study conducted on pregnancy overweight and obese women [74] showed that the abundance of *Collinsella* correlates directly with the circulating levels of insulin, regardless the body mass index (BMI) of the pregnant woman, while it is inversely correlated with the fiber dietary intake. In particular, a low-fiber diet is associated with a gut microbiota that favors the fermentation of lactates. Therefore, it is hypothesized that the composition of the gut microbiota in pregnancy may contribute to the modulation of glucose metabolism.

In a gnotobiotic mouse model, in which the animal colon was colonized by synthetic gut microbiota [75] made up of all commensal bacteria, it was possible to understand the correlation between dietary fiber intake and the integrity of colon mucosal barrier. A reduced intake of fibers leads the gut microbiota to use the muco-glycoproteins produced by the host as a source of nutrients, inducing an alteration of the intestinal mucosa barrier of the colon. This alteration of the colonic mucosa allows the entry of enteric pathogens, such as *Critobacter rodentium*, causing colitis. Therefore, a reduction in fiber intake causes an alteration of the gut microbiota, that induce a dysfunction of the intestinal barrier. This study highlights how a correct intake of fibers can be useful in the management of intestinal pathologies.

## 4. Conclusions

The MD modulates the gut microbiota increasing its diversity and changing the proportion of some bacteria. In fact, a Mediterranean-type dietary pattern is associated with specific characteristics of the microbiota, especially, but not exclusively, in the intestine.

The available evidence suggests that the microbiota of subjects with a Mediterranean-type diet is significantly different from that of subjects with a Western food model. The Mediterranean microbiota would produce more SCFAs that should be able to contribute on the reduction of the risk of both cardiovascular and some tumor pathologies. However, the documentation of the cause-effect relationships between gut microbiota and the risk of these pathologies is incomplete and must necessarily include controlled intervention studies on the microbiota itself, in order to understand the link with dietary patterns.

## Figures and Tables

**Figure 1 nutrients-13-00007-f001:**
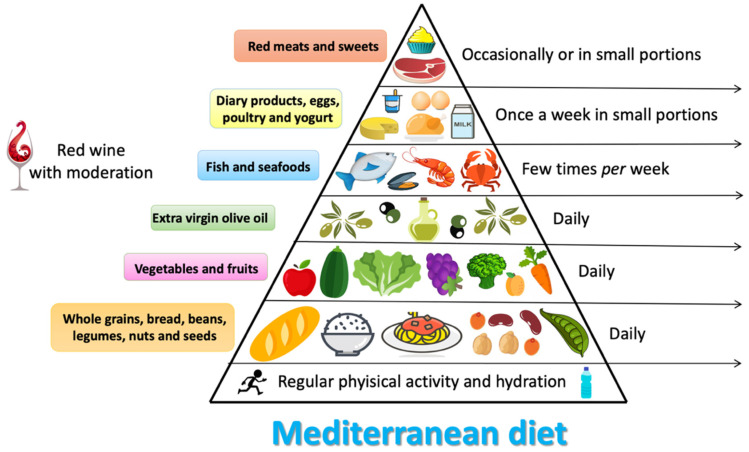
Characteristics of Mediterranean diet.

**Figure 2 nutrients-13-00007-f002:**
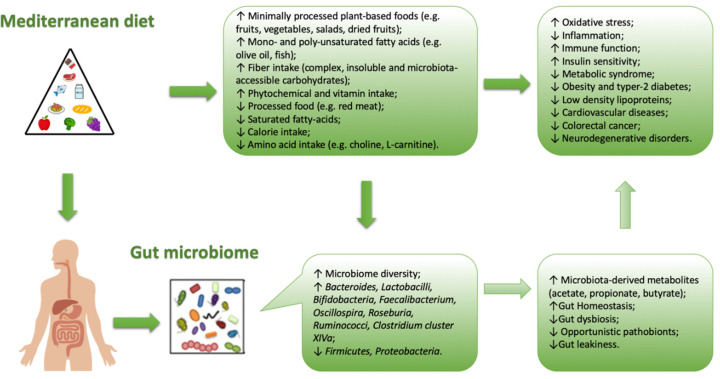
Correlation between Mediterranean diet and intestinal bacterial growth (microbiota).

**Figure 3 nutrients-13-00007-f003:**
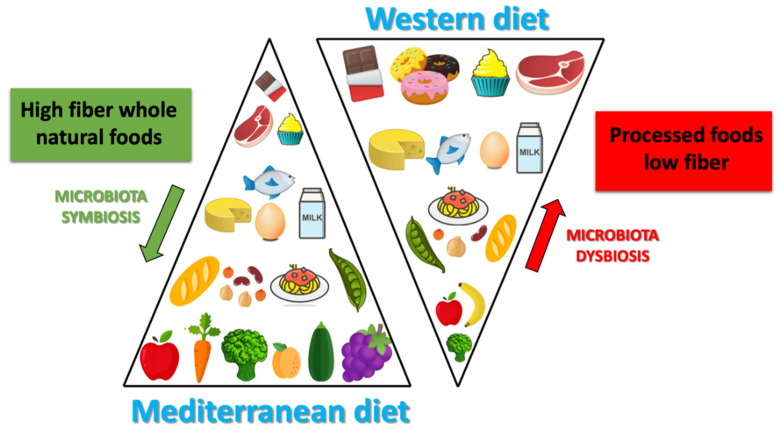
Differences between Mediterranean diet and Western diet.

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
