# Peer review of "Influence of Mediterranean Diet on Human Gut Microbiota"

_nutrients, 2020, doi:10.3390/nu13010007_

Round 1

Reviewer 1 Report

The authors have written a review about the influence of a MedDiet on metabolome. This is an actual field that has to be well explained.

The reviewer requests various corrections to improve the quality of the article:

  • In the abstract section, the results have to be more explained. It seems as a conclusion. In that way, conclusion section has to be resumed.
  • Reference 1 is not explained in the text.
  • References 7-9 are not results from the PREDIMED study. Please, specify the articles from this study that speak about these association. For example: https://www.nejm.org/doi/full/10.1056/NEJMoa1800389.
  • The introduction has many shortcomings. The argument is not new, it talks about well-known studies, some paragraphs are disjointed and the subject of metabolomics is not discussed. This section must be improved. I recommend first speak about what is MedDiet and later it´s effects.
  • Methodology: If the authors speak about the methods used, they should speak how many articules did they find and how many have been refused?
  • Line 116-117- to name the types of bacteria
  • Line 130-131- to define the concep of "prebiotics" or differentiate from the concept of "probiotic".
  • Line 138-140. This paragrah must be improved. It´s difficult to understand as the wester diet appeared at the end of the text. Firmicutes and Bacteroidetes are positive bacteria? Please, explain it.
  • Line 145-149. There is no references. The lower incidence in some types of cancer is due to the microbiota? So what mechanism causes that association to occur. It is a very direct statement and must be well explained. The mechanism are not explained in the article.
  • Figure 3 does not provide any essential information for the reader. The reviewer recommend a table or figure with differences in bacteria between MedDiet and Wester diet.
  • Line 178- to reference Tosti paper in the text.
  • Ref 57-60. These studies relate the effects on the w-3 on microbiota and it association with this patologies?
  • Line 268: If you name numerous studies, tell what studies are them.
  • Lines 277-278. This is in general or due to some patology or diet? If it´s in general, this paragrah shouldn´t be at the end of the manuscrit.
  • Line 280- SCFA is detailed before (line 139)
  • Lines 281-282. This affirmation has to be explained. It´s very general
  • Lines 284-24- The paragraphs do not present a cohesion between them.
  • Conclusion. Is not recommended to add an image in this section.
  • Line 309- It could be interesting to add the type of bacterias that increase with a MedDiet consumption

Author Response

Rome,9th December 2020

Dear Editor,

all the authors would like to thank the Reviewers for their careful reading of the paper.

Please find enclosed the revised version of the manuscript entitled “Influence of Mediterranean diet on human metabolome” to consider for the publication on Nutrients (manuscript No. 987533).

The manuscript has been revised according to all the Reviewers’ comments and all the corrections have been made with “red color” in the manuscript revised version.

Reviewers' #1 comments:

We thank the reviewer for his/her comments.

He/she wrote:

References 7-9 are not results from the PREDIMED study. Please, specify the articles from this study that speak about these association. For example: https://www.nejm.org/doi/full/10.1056/NEJMoa1800389.

We added the suggested reference in the text.

He/she wrote:

Line 116-117- to name the types of bacteria

We added the types of bacteria in the text.

He/she wrote:

Line 130-131- to define the concept of "prebiotics" or differentiate from the concept of "probiotic".

We added the definition of prebiotics.

He/she wrote:

Line 138-140. This paragrah must be improved. It´s difficult to understand as the wester diet appeared at the end of the text. Firmicutes and Bacteroidetes are positive bacteria? Please, explain it.

We amended it according to your suggestion.

He/she wrote:

Line 145-149. There is no references. The lower incidence in some types of cancer is due to the microbiota? So what mechanism causes that association to occur. It is a very direct statement and must be well explained. The mechanism are not explained in the article.

We added the references by [Weng et al.] and [Dai et al.] in the text.

He/she wrote:

Figure 3 does not provide any essential information for the reader. The reviewer recommend a table or figure with differences in bacteria between MedDiet and Wester diet.

We deleted “Figure 3” from the revised paper.

He/she wrote:

Line 178- to reference Tosti paper in the text.

We added the reference of the paper by Tosti et al.

He/she wrote:

Ref 57-60. These studies relate the effects on the w-3 on microbiota and it association with this patologies?

We delete the citation n 57 and replace it with the following:

Impact of omega 3 fatty acids on the gut microbiota. Int J Mol Sci; 2017 Dec 7; 18 (12): 264-5

He/she wrote:

Line 268: If you name numerous studies, tell what studies are them.

We added more papers about this issue (ref. n 60, 70)

He/she wrote:

Lines 277-278. This is in general or due to some patology or diet? If it´s in general, this paragrah shouldn´t be at the end of the manuscrit.

We deleted it.

He/she wrote:

Line 280- SCFA is detailed before (line 139)

We deleted the long form.

He/she wrote:

Lines 281-282. This affirmation has to be explained. It´s very general

We modified the sentence and added some new references.

He/she wrote:

Lines 284-24- The paragraphs do not present a cohesion between them.

We modified them.

He/she wrote:

Conclusion. Is not recommended to add an image in this section.

We removed it

He/she wrote:

Line 309- It could be interesting to add the type of bacterias that increase with a MedDiet consumption

Ok, done

Best regards,

Giuseppe Merra

Reviewer 2 Report

Dear Authors,

Thank you for the opportunity to review your manuscript “Influence of Mediterranean diet on human metabolome”

I would suggest that the tittle needs to be changes to represent your review being on Gut Microbiome in humans.

Please review my further comments below.

Abstract: you may the conclusion ; “The available evidence, suggests that the microbiota of subjects with a Mediterranean-29 type diet is significantly different from that of subjects with a Western food model. The 30 Mediterranean microbiota should be able to contribute on the reduction of the risk of both 31 cardiovascular pathologies and some tumor pathologies.” How have you come to this conclusion??

Introduction

You state : “The Mediterranean diet (MD) [1] has been described for the first time by Keys et al in 1980 [2].” Please note this is not quite correct and the statement needs to be adjusted.

The introduction does not lead he reader into reasons why you are reviewing the Mediterranean diet over other diet patterns? Can you please expand on this further?

 Figure 1 is interesting, did you create this or was it reference form another source, please clarify.

You state that in line 97 : “In this review, we will analyze the interactions between the Mediterranean diet and the human  metabolome response. In fact, we explore the possible modulation of the human metabolome, in response to Mediterranean diet adherence, focusing the attention on EVOO, polyunsaturated fatty acids (PUFA) ω-3 and fiber.” However, you actually referred to human microbiota or gut microbiota.  Please address this issue with the text.

Material and Methods: As you have not followed the PRISMA format for a review it is very difficult to understand why you have chose the respective papers to review and there is a discrepancy within the review and it is not understood why you introduce a non-human primate discussions?  Can you please elaborate on this?

At this point within the manuscript the rationale for the remainder of the manuscript needs a complete review.

There is no indication of what microbiota strains you are considering as beneficial as a result of the Med Diet and why they can make a significant difference to the health of the individual as there is no PICO to substantiate your rationale or discussion.

Author Response

We thank the reviewer for his/her comments.

He/she wrote:

Abstract: you may the conclusion ; “The available evidence, suggests that the microbiota of subjects with a Mediterranean-29 type diet is significantly different from that of subjects with a Western food model. The 30 Mediterranean microbiota should be able to contribute on the reduction of the risk of both 31 cardiovascular pathologies and some tumor pathologies.” How have you come to this conclusion??.

We reviewed the whole abstract section and we rephrased as follows “The available evidence, suggests that the microbiota of subjects that follow a MD is significantly different from subjects that follow a Western food model. In fact, the latter show an increased gut permeability which is responsible of metabolic endotoxemia. For this reason, we can speculate that the microbiota of the subjects following a MD is able to prevent the onset of chronic non-communicable degenerative diseases, including cardiovascular diseases and some types of cancer.”

He/she wrote:

You state : “The Mediterranean diet (MD) [1] has been described for the first time by Keys et al in 1980 [2].” Please note this is not quite correct and the statement needs to be adjusted.

As suggest by the reviewer, we changed the statement as follows: The "concept" of Mediterranean diet (MD) [1] was introduced and characterized for the first time by Ancel Keys, a physiologist, who subsequently studied the effects of eating habits on the incidence of cardiovascular diseases (CD). The results of Keys work have produced the well-known Seven Countries Study”.

He/she wrote:

The introduction does not lead he reader into reasons why you are reviewing the Mediterranean diet over other diet patterns? Can you please expand on this further?

We added and rephrased the section as follows: “The role of diet in the gut microbiota modulation has been widely recognized [2,3]. De Filippis et al [4] demonstrated that a Mediterranean-type diet, characterized by a high content of plant-based food, exert a beneficial role in the gut microbiota composition. In particular, subjects who eat a higher proportion of plant-based foods show a higher percentage of short chain fatty acids (SCFAs) and of fiber-degrading bacteria in the feces. While, subjects with poor adherence to the Mediterranean diet have a higher concentration of trimethylamine N-oxide (TMAO) in the urine.

In a subsequent study Mitsou et al [5] confirmed the positive impact of the Mediterranean diet on the intestinal microbiota profile. In fact, the subjects who had a higher adherence to the Mediterranean diet, monitored with the MedDietScore, had a lower presence of E. coli and an increased total presence of bacteria, a higher Bifidobacteria/ E. coli ratio and an increased prevalence of C. albicans and SCFAs.

Recently, Garcia-Mantrana et al [6] pointed out that a higher adherence to the Mediterranean diet was characterized by an enhancement of Bifidobacteria and a higher percentage of SCFAs. Therefore, MD had a positive impact on the gut microbiota profile, on its Alpha-diversity and on its metabolic activities.

In this review, we will analyze the impact of Mediterranean diet on the human metabolome response. In fact, we explore the possible modulation of the human metabolome, in response to Mediterranean diet adherence, focusing the attention on MD typical elements such as EVOO, polyunsaturated fatty acids (PUFA) ω-3 and fiber.”

He/she wrote:

Figure 1 is interesting, did you create this or was it reference form another source, please clarify.

Figure 1 was created in original form by the authors. In fact, it refers to the classic food pyramid, commonly known.

He/she wrote:

You state that in line 97 : “In this review, we will analyze the interactions between the Mediterranean diet and the human  metabolome response. In fact, we explore the possible modulation of the human metabolome, in response to Mediterranean diet adherence, focusing the attention on EVOO, polyunsaturated fatty acids (PUFA) ω-3 and fiber.” However, you actually referred to human microbiota or gut microbiota.  Please address this issue with the text.

We corrected in the whole manuscript the sentence “human microbiota” with “gut microbiota”.

He/she wrote:

Material and Methods: As you have not followed the PRISMA format for a review it is very difficult to understand why you have chose the respective papers to review and there is a discrepancy within the review and it is not understood why you introduce a non-human primate discussions?  Can you please elaborate on this?

OK, we have eliminated rightly the discussion on primate

He/she wrote:

At this point within the manuscript the rationale for the remainder of the manuscript needs a complete review.

OK, we made a complete renovation of paragraph

He/she wrote:

There is no indication of what microbiota strains you are considering as beneficial as a result of the Med Diet and why they can make a significant difference to the health of the individual as there is no PICO to substantiate your rationale or discussion.

We better explained the modification of gut microbiota composition in course of some pathological conditions: “The phyla mainly present in the gut microbiota are Firmicutes and Bacteroidetes, which represent about 90% of the intestinal bacterial flora[7].

The main genera that make up the Firmicutes phylum are Clostridium, Enterococcus, Lactobacillus and Ruminococcus. While the main constructors of the phylum Bacteroidetes are Prevotella and Bacteroides. Many pathological conditions are characterized by a dysbiosis of the intestinal microbiota [8]. For example, in presence of CKD, the alteration of the gut microbiota composition is characterized by the prevalence of Enterobacteriaceae, Lachnospiraceae, Ruminococcaceae and a decrease in Bifidobacteriaceae, Bacteroidaceae, Lactobacillacee and Prevotellaceae [8]. Moreover, it has recently been shown that dysbiosis is directly correlated with the onset of colorectal cancer (CRC). In particular, the presence of some bacterial species such as Fusobacterium nucleatum, Peptostreptococcus anaerobius and enterotoxigenic, Bacteroides fragilis can cause the proliferation of tumor cells, induce an inflammatory state and damage to the DNA with consequent impact on the onset of CRC. While, the reduction of probiotic bacteria such as Lachnospiraceae, Bifidobacterium animalis and Streptococcus thermophilus, that seem to exert a protective action in CRC, is present in patients affected by this neoplastic pathology [9].”

The whole manuscript has been revised for the English language.

Best regards,

Giuseppe Merra

Round 2

Reviewer 2 Report

Dear Authors, 

My only further suggestion would be to change the title of your manuscript to " Influence of Mediterranean Diet on human gut Microbiota" as this is what you are now discussing in your manuscript.

All other corrections are acceptable.

Author Response

Prof. Giuseppe Merra

Rome,16th December 2020

Dear Editor,

all the authors would like to thank the Reviewers for their new careful reading of the paper.

Please find enclosed the revised version of the manuscript already entitled “Influence of Mediterranean diet on human metabolome” now corrected on just suggestion of Reviewer 2 in: "Influence of Mediterranean Diet on human gut Microbiota" to consider for the publication on Nutrients (manuscript No. 987533).

The manuscript has been revised according to all the Reviewers’ comments and all the corrections have been made with “red color” in the manuscript revised version.

Best regards,

Giuseppe Merra